# OpenReview forum: "What Generative Search Engines Like and How to Optimize Web Content Cooperatively"
_ICLR.cc/2026/Conference — ICLR 2026 Poster_

### Official Review · Reviewer_WpqU · 2025-10-29

**Soundness:** 3
**Presentation:** 3
**Contribution:** 2
**Rating:** 6
**Confidence:** 4

**Summary:**

In this paper, the authors address the task of Generative Engine Optimization (GEO) and introduce a framework called AutoGEO, which aims to rewrite web content to improve its visibility on generative search engines. To achieve this, large language models (LLMs) are employed to extract preference rules by analyzing documents with contrasting visibility levels. The authors then propose two models to learn these rules: a prompt-based model and another trained using GRPO, which optimizes a reward composed of three components. Experiments conducted on three datasets demonstrate that AutoGEO achieves significant performance gains over baseline methods.

**Strengths:**

- S1: The problem addressed in this work is realistic and timely, given the increasing role of generative search engines in content discovery. The paper offers a practical and effective solution that aligns well with how web content creators may optimize for LLM-based visibility.

- S2: The proposed two-stage framework is conceptually clear and straightforward to follow. The writing is clean and the overall presentation is well-organized, making the methods understandable and likely reproducible. The implementation choices are also reasonable and grounded.

- S3: The experimental evaluation is comprehensive. The authors benchmark across multiple datasets from different domains and evaluate performance on several LLMs. The transferability analysis showing how rule sets may generalize across engines or domains is particularly interesting and contributes useful insight into how stable such optimization strategies may be in practice.

- S4: The results are strong and consistently show improvements over baselines.

**Weaknesses:**

- W1: The rule extraction process heavily relies on LLM-generated rationales, yet there is limited qualitative or human-driven analysis of the extracted rules themselves.

- W2: There is no ablation study breaking down the effects of each stage of the rule extraction pipeline. Since the rules are represented in free-form text, there is a reasonable possibility that multiple rules encode highly similar behaviors or overlapping preferences. A redundancy or compression analysis (e.g., clustering or semantic similarity grouping) would help clarify how many distinct principles are truly being captured.

- W3: While the method demonstrates that individual documents can be optimized for visibility, it is unclear how the system behaves under global optimization—i.e., if many or all documents adopt similar optimized styles. A small-scale controlled experiment (e.g., optimizing all documents within a subset and measuring relative visibility or stylistic homogenization) would provide insight into potential equilibrium

**Questions:**

- Q1: Are there instances where the rewritten documents introduce hallucinated or unsupported information?

---

> ### Author Response · Authors · 2025-11-21
>
> > ### **W1: The rule extraction process heavily relies on LLM-generated rationales, yet there is limited qualitative or human-driven analysis of the extracted rules themselves.**
>
> Thank you for pointing this out. We include a human evaluation (`General Response`) on rule quality shows that 95.47% of rules are rated medium or high across all quality criteria, and 76.9% are meaningfully aligned with the GE domain. In addition, our paper includes analyses: `Section 5.3` examines cross-engine rule patterns using human-annotated keypoints at the rule level, `Appendix C` provides the full rule lists with keyword annotations for comparison and inspection, and `Appendix M` presents case studies illustrating how rules guide document rewriting. We have also added more detailed domain-specific rule case studies in the `Supplementary Material`. We hope these analyses together offer a clearer picture of the rationality of the extracted rules.
>
> ---
>
> > ### **W2.1: There is no ablation study breaking down the effects of each stage of the rule extraction pipeline.**
>
> Thank you for the suggestion. We have added a new ablation study on Gemini generative engine on Researchy GEO dataset that isolates the contribution of each component in the rule extraction pipeline (Explainer, Extractor, Merger, Filter). For each variant, we remove exactly one component while keeping all remaining stages unchanged.
>
> | Variant        | Word | Pos   | Overall  |
> |----------------|----------|----------|-----------|
> | w/o Explainer  | 42.72    | 42.63    | 42.91     |
> | w/o Extractor  |    41.25      |    41.20      |     41.78      |
> | w/o Merger     | 39.71    | 39.89    | 40.28     |
> | w/o Filter     | 42.85    | 43.35    | 43.71     |
> | **AutoGEO$\_\text{API}$**| **42.87**| **43.53**| **43.76** |
>
> **Observations:** Removing Extractor or Merger leads to the largest performance drops because irrelevant information enters the rule set. Removing Explainer or Filter has only minor impact: insights can still be inferred from document pairs without Explainer, and Filter only removes a few weak rules. The full pipeline achieves the best performance, though partial pipelines still yield meaningful gains. Overall, the results show that GE preference modeling is driven primarily by data-driven signals rather than any single pipeline stage. We will include this ablation in the revision.

---

> ### Author Response · Authors · 2025-11-21
>
> > ### **W2.2: Since the rules are represented in free-form text, there is a reasonable possibility that multiple rules encode highly similar behaviors or overlapping preferences. A redundancy or compression analysis (e.g., clustering or semantic similarity grouping) would help clarify how many distinct principles are truly being captured.**
>
> Thank you for raising this point. Our pipeline already mitigates redundancy explicitly.
>
> **1. The Merger stage removes redundant or overlapping rules.**
> As described in `Section 3.1`, the Merger step performs semantic clustering of candidate rules and summarizes each cluster into one representative rule.  This directly addresses the reviewer's concern. Below we show the token and rule counts at each stage for the largest dataset, Researchy GEO:
>
> |   Stage   | Before Explainer | Before Extractor | Before Merger | Before Filter | Final |
> |:-----------|:--------|:--------|:-----------|:--------|:---------|
> | \#Token | 19,249,056 | 6,892,988 | 1,195,779 | 182 | 163 |
> | \#Rule | -  | -  | 71,403 | 16 | 15 |
>
> The Merger compresses 71,403 insights to 16 rules, demonstrating substantial redundancy removal.
>
> **2. Human inspection confirms distinct semantics.**
> In `Section 5.3` and `Appendix C`, we manually check the merged rules using human-annotated keywords. Annotators do not find cases where two rules encode the same preference, supporting that the final rules are semantically distinct.
>
> Taken together, the clustering-based Merger stage and the human validation show that the final rule sets are semantically distinct and capture different preference rules of generative engines.
>
> ---
> > ### **W3: While the method demonstrates that individual documents can be optimized for visibility, it is unclear how the system behaves under global optimization—i.e., if many or all documents adopt similar optimized styles. A small-scale controlled experiment (e.g., optimizing all documents within a subset and measuring relative visibility or stylistic homogenization) would provide insight into potential equilibrium**
>
> Thank you for the thoughtful suggestion. To directly address this question, we conducted an additional **global optimization experiment** on the Researchy GEO dataset using the Gemini generative engine.
>
> We compare two settings:
> - **Target Optimization:** Only the target document is rewritten using AutoGEO (the default setting in the paper).
> - **Global Optimization:** All documents in the evaluation subset are rewritten using the same rule set, thereby simulating a scenario where many authors adopt similar GEO-optimized writing practices.
>
> | Method                                   | Word ↑ | Pos ↑  | Overall ↑ | KPC ↓  | KPR ↑   | Precision ↑ | Recall ↑ | Clarity ↑ | Insight ↑ |
> |---------------------------------|--------|--------|-----------|--------|---------|-------------|----------|-----------|-----------|
> | Vanilla                                  | 20.11  | 20.13  | 20.18     | 0.27   | 40.33   | 96.05       | 99.22    | 60.10     | 51.07     |
> | AutoGEO$\_\text{API}$ (Target)        | **42.87** | **43.53** | **43.76** | **0.24** | 42.40   | 97.02       | 99.17    | 61.97     | 53.79     |
> | AutoGEO$\_\text{API}$ (Global)    | 19.34  | 19.34  | 19.23     | 0.32   | **45.76** | **98.97**   | 99.33    | **64.15** | **59.38** |
> | AutoGEO$\_\text{Mini}$ (Target)       | 37.50  | 38.37  | 38.53     | 0.34   | 40.33   | 96.89       | **99.45** | 61.48     | 52.67     |
> | AutoGEO$\_\text{Mini}$ (Global)   | 19.64  | 19.57  | 19.65     | 0.32   | 43.40   | 98.17       | 99.21    | 61.45     | 57.39     |
>
> This global optimization experiment shows that when all documents adopt the same AutoGEO-optimized rules, the relative **visibility advantage of individual documents decreases**, with Word, Pos, and Overall approaching Vanilla levels. At the same time, metrics related to generative engine utility (KPR, Precision, Clarity, Insight) improve, reflecting more uniform content and stylistic homogenization. This suggests that while individual advantage is reduced under global adoption, AutoGEO remains **cooperative with the engine** and does not generate hallucinated or unsupported information. We will include this interesting experiment in the revision.

---

> ### Author Response · Authors · 2025-11-21
>
> > ### **Q1: Are there instances where the rewritten documents introduce hallucinated or unsupported information?**
>
> Thank you for raising this concern. We have carefully examined whether AutoGEO introduces hallucinated or unsupported content and provide multiple lines of evidence showing that this is not the case.
>
> 1. **Prompt and rules unrelate to adding new facts.**
> As detailed in `Appendix B` (rewriting prompt template) and `Appendix C` (full rule sets), none of the rules encourage introducing external or unverifiable information. The rules focus exclusively on structural visibility cues, such as clarity, specificity, and organization, without prompting content augmentation beyond the original document.
>
> 2. **Generative engine utility does not decrease.**
> Across all generative engines, GEU, which evaluates the performance of the generative engines, remains stable as shown in `Table 2`. If hallucinated or unsupported content were introduced, GEU would likely degrade as visibility increases. The stable GEU strongly suggests that rewritten documents preserve factual consistency.
>
> 3. **Case study verification.**
> `Appendix M` already presents before-and-after rewriting case studies. We have further expanded this analysis with two additional randomly sampled cases from the Researchy GEO dataset (more cases and full document content are provided in the `Supplementary Material`). For each case, we compare the original document with its rewritten counterpart, using keypoints extracted by the LLM, and observe no instances of hallucinated or unsupported content. The rewrites primarily reorganize and clarify existing information rather than introducing new material.
>
> 4. **Future enhancements are feasible.**
> While our current design already avoids hallucination, further improvements are feasible, such as:
> - more explicit factuality-preserving constraints in prompting, or
> - reward modeling during training to penalize unsupported content.  Recent work (e.g., RePro [1]) shows that targeted training can significantly reduce hallucination, suggesting a promising direction for AutoGEO.
>
> **Summary:** Through prompt template design, rule constraints, GEU evaluation, and case analysis, we find no evidence that AutoGEO introduces hallucinated or unsupported information.
>
> [1] Yu, Zichun, and Chenyan Xiong. "RePro: Training Language Models to Faithfully Recycle the Web for Pretraining." arXiv preprint arXiv:2510.10681 (2025).
>
> ---
>
> ### Document 1
>
> Core content
>
> ```text
> The document explains how India plans to improve road safety by deploying five traffic technologies—LIDAR guns, speed displays, speed governors, variable message signs, and inductive loops—within an integrated traffic management system.
> ```
>
> Key points (coverage 100%, all key points in the rewritten document appear in the original)
>
> ```text
> 1. Both versions describe the same five technologies and link them to road-safety and enforcement goals in India.
> 2. Both explain how LIDAR guns and speed displays detect speeding and provide evidence or feedback to drivers.
> 3. Both present speed governors in commercial vehicles as devices that cap maximum speed by restricting fuel or air supply.
> 4. Both highlight variable message signs and inductive loops as tools for real-time traffic monitoring and signal control.
> 5. Both frame the overall approach as part of an Integrated Traffic Management System rather than isolated gadgets.
> ```
>
>
> ### Document 2
>
> Core content
>
> ```text
> The document analyzes how terrorism indirectly disrupts business supply chains through tighter security policies, customs delays, higher costs, and how initiatives like C-TPAT help firms manage these risks.
> ```
>
> Key points (coverage 100%, all key points in the rewritten document appear in the original)
>
> ```text
> 1. Both versions emphasize that terrorism mainly affects supply chains indirectly via fear and government security responses rather than only physical damage.
> 2. Both describe post-9/11 laws and regulations that increase inspections and security at ports, airports, and other transport hubs.
> 3. Both highlight the resulting delays, congestion fees, and reduced efficiency for shippers and manufacturers.
> 4. Both explain the purpose and basic operation of C-TPAT as a voluntary partnership that rewards certified "low-risk" firms with expedited processing and fewer inspections.
> 5. Both stress that security-induced costs can, in some cases, exceed the direct economic losses from the terrorist event itself.
> ```

---

> ### Comment · Reviewer_WpqU · 2025-11-28
>
> Thanks to the authors for the additional experiments, which have addressed all my concerns. I will raise the score accordingly, after the funtionality is enabled ...

---

> > ### Author Response · Authors · 2025-11-28
> >
> > We sincerely appreciate your positive feedback and the time and effort you have dedicated to evaluating our work.

---

### Official Review · Reviewer_CXWv · 2025-11-01

**Soundness:** 2
**Presentation:** 2
**Contribution:** 2
**Rating:** 4
**Confidence:** 2

**Summary:**

This paper introduces AutoGEO, a new GEO method for generative engines that generates responses based on retrieved contents.
AutoGEO leverages LLMs to extract preference rules from a generative engine and rewrites web contents based on these preference rules such that corresponding contents gain more traction in generative engines' generated responses.
Two strategies, including $AutoGEO_{API}$ and $AutoGEO_{Mini}$, are proposed.
The former leverages prompt engineering on LLMs with frozen parameters and the latter introduces a compact model fine-tuned using SFT and RL for cost-efficient deployment.
Experiments on several benchmarks demonstrates the effectiveness of AutoGEO.

**Strengths:**

1. Writing is good and the paper is easy to follow.

2. This paper focuses on GEO, which is an important and practical problem. Since generative engines become more and more popular, understanding and resolving GEO is crucial, just as the status of SEO in conventional search engine. The proposed method will be beneficial for future work in this area.

3. Experimental evaluation is extensive. This paper consider not only existing GEO benchmark but also constructing two new benchmarks, which enhances the diversity and robustness of the evaluation of AutoGEO. Besides, detailed analysis on GEO and GEU and ablation study strengthens the effectiveness of AutoGEO.

**Weaknesses:**

1. The novelty is limited. AutoGEO seems a direct application of existing techniques with little novelty in learning paradigm and theoretical understanding. The process of using a powerful LLM to explain, extract, and merge preference rules is essentially a form of automated data labeling and summarization, and using these extracted rules as prompts for a rewriting model (i.e., $AutoGEO_{API}$) is standard in-context learning. Besides, using rules as reward signals in RL for fine-tuning LLMs (i.e., $AutoGEO_{Mini}$) is also a direct application of existing methods. It develops a competent application system which achieves impressive GEO performance, but does not make a foundational machine learning contribution.

2. The analysis on cost is limited. This paper argues the cost efficiency of $AutoGEO_{Mini}$ compared to $AutoGEO_{Mini}$ (i.e., ~0.071x), but ignores the cost of rule extraction phase, which seems to be substantial and dominant. For evaluating a practical application, the cost of the entire process should be analyzed.

3. Analysis on preference rules is limited. The rule extraction pipeline obtains the final rule set through components like Explainer, Extractor, Merger, and Filter. However, clear metrics or methods to measure the accuracy, effectiveness or rationality of these rules themselves are missed and it is validated in an indirect way via downstream tasks. This approach only shows that rules are useful, but does not verify that a rule reflects the true preference of generative engines instead of noise from the dataset or inherent bias of LLMs.

**Questions:**

In summary, this paper develops an effective application for resolving GEO problem, which is an important application domain. However, the algorithmic or theoretical contribution is limited. So I think this paper is more aligned with the field of applied data science like KDD, WWW and SIGIR.

This paper can be further improved, including:
1. Adding theoretical analysis, e.g., discussing the condition under which the proposed method is guaranteed to work
2. Adding analysis of the effectiveness and rationality of preference rules

---

> ### Author Response · Authors · 2025-11-21
>
> > ### **W1: The novelty is limited. AutoGEO seems a direct application of existing techniques with little novelty in learning paradigm and theoretical understanding. The process of using a powerful LLM to explain, extract, and merge preference rules is essentially a form of automated data labeling and summarization, and using these extracted rules as prompts for a rewriting model (i.e., AutoGEO$\_\text{API}$) is standard in-context learning. Besides, using rules as reward signals in RL for fine-tuning LLMs (i.e., AutoGEO$\_\text{Mini}$) is also a direct application of existing methods. It develops a competent application system which achieves impressive GEO performance, but does not make a foundational machine learning contribution.**
>
> Thank you for the thoughtful comment. We respectfully clarify that AutoGEO is not a direct application of existing techniques but introduces new designs, new insights, and a new optimization notion.
>
> **1.** AutoGEO includes **GEO-specific** methodological innovations. Prior rule-extraction methods operate at small, instance-level scales, whereas GEO requires corpus-level visibility analysis. AutoGEO provides the necessary aggregation and hierarchical merging mechanisms, and both AutoGEO$\_\text{API}$  and AutoGEO$\_\text{Mini}$ operationalize these rules to achieve a GEO–GEU trade-off that generic prompting and standard RLHF cannot support.
>
> **2.** Our data-driven analysis produces the first structured and empirically validated rule set that explains generative-engine visibility preferences. These rules are stable and interpretable, **offering insights** that can guide future research on generative search systems.
>
> **3.** We introduce a **cooperative GEO framework** that jointly optimizes GEO and maintains GEU, ensuring visibility gains without harming usefulness or alignment. This GEO–GEU formulation is new and shifts the problem from "pure optimization" to building win–win content–engine interactions, which is essential for the emerging generative-engine ecosystem. As generative engines become the default interface for accessing information, modeling how content can improve visibility while preserving generative engine utility becomes an important and emerging research direction. Our framework takes a first step toward systematically formalizing this interaction and provides a principled foundation for future work on cooperative content–engine optimization.
>
> **In summary**, AutoGEO contributes (i) customized methodological designs, (ii) new empirical understanding of generative-engine preferences, and (iii) a novel cooperative framework, together forming a substantive conceptual and technical step beyond directly applying existing methods.

---

> ### Author Response · Authors · 2025-11-21
>
> > ### **W2: The analysis on cost is limited. This paper argues the cost efficiency of compared to  (i.e., ~0.071x), but ignores the cost of rule extraction phase, which seems to be substantial and dominant. For evaluating a practical application, the cost of the entire process should be analyzed.**
>
> We appreciate the reviewer's observation and list the rule-extraction cost.
>
> **1. We now include the rule-extraction cost.**
> The table below presents the costs for Explainer, Extractor, Merger, and Filter across all three datasets' training sets, including both total and per-sample values, for the Gemini generative engine.
>
> | Dataset       | \#Sample | Explainer | Extractor | Merger | Filter | Total | Avg.   |
> |:--------------|:------------|:----------|:----------|:-------|:-------|:------|:-------|
> | E-commerce   | 1,667        | 0.1042    | 0.0466    | 1.145  | 0.0003 | 1.428 | 0.0009 |
> | GEO-Bench    | 8,000        | 0.3021    | 0.0816    | 1.821  | 0.0004 | 2.726 | 0.0003 |
> | Researchy GEO| 10,000       | 0.3350    | 0.1230    | 2.504  | 0.0005 | 3.635 | 0.0004 |
>
> **2. The extraction cost is incurred only once and becomes negligible when amortized over many documents.**
> We run the extraction pipeline **one time** for each generative engine. The resulting rule set is **reused** for GEO models. So in practice, users rewrite large numbers of documents, which quickly dilutes the one-time extraction cost:
> - AutoGEO$\_\text{API}$: ~99.7% of total cost for 1k documents, ~24.8% for 1M, ~0.03% for 1B.
> - AutoGEO$\_\text{Mini}$: ~70.8% for 1k, ~0.24% for 1M, ~0.00024% for 1B.
>
> Thus, the dominant cost in practical applications is inference, not extraction.
>
> **3. The 0.0071× number is about inference efficiency.**
> Both models share the same one-time rule extraction, so including it does not change their relative efficiency. The 0.0071× ratio as explained in `Appendix E.5` refers specifically to inference-time cost on GEO-Bench test set:
> - AutoGEO$\_\text{API}$: Gemini-2.5-Pro, \\$1.25 per million input tokens, \\$10 per million output tokens.
> - AutoGEO$\_\text{Mini}$: Qwen3-1.7B, on NVIDIA A6000 Ada GPU, \\$0.75/hour.
>
> The total AutoGEO cost per dataset (extraction + inference) is summarized below:
> | Dataset | \#Training Sample |  Rule Extraction | \#Test Sample | AutoGEO$\_\text{API}$ | AutoGEO$\_\text{Mini}$ |
> |:--------------|:------------|:----------|:-------|:-------|:------|
> | E-commerce   | 1,667        | 1.428  | 416 | 0.711 | 0.0053 |
> | GEO-Bench    | 8,000        | 2.726  | 1,000 | 1.512 | 0.0122 |
> | Researchy GEO| 10,000       | 3.635  | 1,000 | 1.498 | 0.0106 |
>
> We will update the paper to explicitly distinguish extraction vs inference costs and include the full extraction-cost table for transparency.
>
> ---

---

> ### Author Response · Authors · 2025-11-21
>
> > ### **W3.1: Analysis on preference rules is limited.**
>
> Thank you for the comment. We add human-driven analysis of their quality, as shown in `General Response`, showing that 95.47% of rules score medium or high on all quality criteria, and 76.93% are meaningfully aligned with GE preferences rather than domain-irrelevant noise. Besides, below we summarize our multi-faceted analysis to clarify the stability, effectiveness, and interpretability of the extracted preference rules in our paper.
>
> **1. What the rules look like across models and domains (`Sec. 5.3`).**
> We compare rules extracted from three LLMs and three datasets. We find both shared patterns (e.g., clarity, structure, factual grounding) and domain-specific rules (e.g., technical depth for Researchy-GEO, product details for E-commerce).  This shows that our method captures stable preferences as well as meaningful domain differences.
>
> **2. Which rules matter the most (`Sec. 5.4`).**
> Our ablation studies show that each single rule helps GEO performance. This means the rules are not noise: each provides useful guidance for visibility.
>
> **3. Full rule sets for transparency (`Appendix C`).**
> We provide the complete rule sets with keyword annotations, enabling direct inspection and comparison of both shared and domain-specific rules.
>
> **4. How rules affect actual generations (`Appendix M`).**
> Case studies illustrate how applying the rules systematically modifies document content to improve visibility.
>
> Together, both quantitative (ablation, GEO score improvements) and qualitative (human study, case studies, human-inspectable rules) evidence support the reliability and effectiveness of preference rules.
>
> ---
>
> > ### **W3.2: The rule extraction pipeline obtains the final rule set through components like Explainer, Extractor, Merger, and Filter. However, Clear metrics or methods to measure the accuracy, effectiveness or rationality of these rules themselves are missed and it is validated in an indirect way via downstream tasks**
>
> Thank you for the question. As described in `Section 4`, we view the utility of the rules—their ability to improve real-world empirical results (GEU)—as the ultimate evaluation. Therefore, downstream GE performance serves as the final assessment. We also agree that intermediate rule-level metrics are useful, and we report GEO scores to measure rule fidelity to GE preferences.
>
> **1. GEO directly measures whether the rules achieve their intended effect.**
> Because the rules are designed to improve GEO, their quality is most directly reflected in the resulting GEO gains. In this sense, the objective itself serves as the most aligned and meaningful measure of rule effectiveness.
>
> **2. Complementary analyses support rule validity.**
> Beyond GEO scores, we evaluate rules through: `Section 5.3` examines cross-engine rule patterns; `Section 5.4` shows ablations confirming each rule contributes to visibility improvement; `Appendix C` provides full rule lists with keyword annotations; `Appendix M` offers case studies illustrating rule-driven edits.
>
> In summary, GEO offers a faithful evaluation of rules' effectiveness, and the complementary analyses further validate their accuracy, coherence, and practical utility.

---

> ### Author Response · Authors · 2025-11-21
>
> > ### **W3.3: This approach only shows that rules are useful, but does not verify that a rule reflects the true preference of generative engines instead of noise from the dataset or inherent bias of LLMs.**
>
> Thank you for the thoughtful question. The following results demonstrate that the extracted rules reflect true GE preferences and are easy to inspect for existing noise or LLM bias.
>
> 1. Human evaluation (`General Response`) shows that 95.47% of rules score medium or high on all quality criteria, and 76.93% are meaningfully aligned with GE preferences rather than domain-irrelevant noise. Additionally, our rule-level human keyword annotation in `Section 5.3` and following `case studies` (more cases are provided in `Supplementary Material`) reveal shared patterns and clear domain-specific preferences across three LLMs and three datasets, consistent with known GE behaviors.
>
> 2. Ablation studies (`Section 5.4`) show that each individual rule contributes to GEO improvement across datasets. Random noise cannot consistently boost visibility; only rules aligned with real GE behavior do.
>
> 3. `Appendix C` provides full rules with annotated keypoints. The rule set is small (fewer than 20 rules) and easy to inspect, enabling direct verification of their meaningfulness.
>
> **Rule Cases of Different Domains**
>
> (1) In-Depth (Research)
>
> ```text
> Provide explanatory depth by clarifying underlying causes, mechanisms, and context ('how' and 'why').
> ```
>
> (2) Conclusion First (Research)
>
> ```text
> State the key conclusion at the beginning of the document.
> ```
>
> (3) Balanced View (Research)
>
> ```text
> Present a balanced perspective on complex topics, acknowledging multiple significant viewpoints or counter-arguments.
> ```
>
> (4) Pros & Cons Rec (E-commerce)
>
> ```text
> Justify recommendations and claims with clear reasoning, context, or comparative analysis like pros and cons.
> ```
>
> (5) Production Details (E-commerce)
>
> ```text
> Provide specific, verifiable details such as names, model numbers, technical specifications, and quantifiable data..
> ```

---

> ### Author Response · Authors · 2025-11-21
>
> > ### **Q1: Adding theoretical analysis, e.g., discussing the condition under which the proposed method is guaranteed to work**
>
> Thank you for the suggestion. We note that this work is application-driven, and our contribution lies in designing a practical method for GEO rather than in theoretical analysis.
>
> Providing formal guarantees would require strong assumptions about the internal scoring and given that theory for LLMs is still in its infancy, such guarantees are currently infeasible for realistic settings. Instead, we demonstrate the robustness and effectiveness of our approach through extensive experiments. Across three datasets (E-commerce, GEO-Bench, Researchy-GEO) and multiple LLMs (Gemini, Claude, GPT), AutoGEO consistently extracts coherent rule sets, improves GEO, and maintains downstream GE performance. This empirical evidence confirms that the method works reliably under diverse real-world conditions.
>
> ---
>
> > ### **Q2: Analysis of the effectiveness and rationality of preference rules**
>
> Thank you for the insightful question. The effectiveness of the rules is already reflected in our main experiments (`Sec. 5`): our rule-grounded GEO models significantly improve GEO while maintaining GEU, and per-rule ablations show that any single rule consistently enhances GEO performance. Besides, we add human evaluation (`General Response`) on rule rationality shows that 95.47% of rules score medium or high on all quality criteria, and 76.93% are meaningfully aligned with GE preferences rather than domain-irrelevant noise, supporting the rules' rationality.

---

### Official Review · Reviewer_YwZa · 2025-11-01

**Soundness:** 3
**Presentation:** 4
**Contribution:** 3
**Rating:** 8
**Confidence:** 4

**Summary:**

This study aims to optimize web documents so that their content gains higher visibility. The key idea is to optimize the utilization of retrieved information via a generative engine. To achieve such a goal, a training-free framework and a RL-based training approach are proposed, which aims to extract preference rules for generative engine optimization in a plug-in or an automatic way. The experimental results based on common metrics and the designed evaluation setting demonstrate the effectiveness. Overall, this is an interesting paper that offers an important perspective on the future of search.

**Strengths:**

1. The target problem is important and interesting. What is the search engine like in the era of LLM and how to leverage the LLM to integrate with search paradigm is a timely topic. This study provides a new perspective for the community to facilitate our thoughts.

2. The proposed methods are effective, which include two different versions: plug-in one and an automatic one with training. The designed framework is reasonable and systematic.

3. Experimental results are good and a series of analyses are provided to support the claim of this study.

**Weaknesses:**

1. The motivation/illustration of the proposed search paradigm could be made more clear by comparing its differences with existing search paradigms, e.g., generative IR model, LLM-based dense retriever, LLM-based unified retriever and generator, and vanilla RAG. A systematic comparison could enhance the contribution in terms of the perspective of "What Generative Search Engines Like".

2. Similar to point 1, a task definition could be desirable before describing the methodology. The defined scenarios can connect with the used downstream datasets and evaluation settings.

3. The involved compared paradigms in point 1 could be included in the related work to better illustrate the connection and differences compared to existing search paradigms. Please see my question below as another enhanced aspect.

**Questions:**

1. What is the difference (and their optimization goal) between the generative engine (as introduced in related work) and the unified framework that aims to unify the retrieval and generation tasks using a single LLM as backbone (e.g., GRIT [1], UniConv [2])? It would be better to illustrate them in the related work, as the paper also wants to provide a perspective of "What Generative Search Engines Like"

[1] Generative representational instruction tuning.

[2] Uniconv: Unifying retrieval and response generation for large language models in conversations.

---

> ### Author Response · Authors · 2025-11-21
>
> > ### **W1:** The motivation/illustration of the proposed search paradigm could be made more clear by comparing its differences with existing search paradigms, e.g., generative IR model, LLM-based dense retriever, LLM-based unified retriever and generator, and vanilla RAG. A systematic comparison could enhance the contribution in terms of the perspective of "What Generative Search Engines Like".
> > ### **W3:** The involved compared paradigms in point 1 could be included in the related work to better illustrate the connection and differences compared to existing search paradigms. Please see my question below as another enhanced aspect.
> > ### **Q1:** What is the difference (and their optimization goal) between the generative engine (as introduced in related work) and the unified framework that aims to unify the retrieval and generation tasks using a single LLM as backbone (e.g., GRIT [1], UniConv [2])? It would be better to illustrate them in the related work, as the paper also wants to provide a perspective of "What Generative Search Engines Like"
>
> Thank you for the helpful comment. We clarify that **our work does not propose a new search paradigm**. The phrase "What Generative Search Engines Like" refers to the **content preferences exhibited by existing Generative Engines (GEs)**.
>
> **1. What counts as a Generative Engine**
> Following prior work [1], we use "Generative Engine (GE)" as a behavior-level concept: any system whose **final output is an LLM-generated, synthesized response grounded in retrieved evidence** qualifies as a GE. Under this definition, GEs are products that often use these techniques:
> - generative IR models,
> - LLM-based dense retrievers,
> - unified retriever–generator models such as GRIT and UniConv
> - standard RAG pipelines.
>
> Our work therefore studies the preference behavior exhibited by these existing GE forms, rather than proposing a new search paradigm.
>
> **2. Contribution of this work**
> Given this clarified definition, our contribution is:
> - to provide a systematic analysis of **what content GEs prefer to surface**, and
> - to develop an automated method to **extract and leverage these preferences** to improve document visibility.
>
> We will revise the paper to make these connections explicit and include a clearer comparison in the related work to avoid confusion.
>
> [1] Aggarwal, Pranjal, et al. "Geo: Generative engine optimization." Proceedings of the 30th ACM SIGKDD Conference on Knowledge Discovery and Data Mining. 2024.
>
> ---
>
> > ### **W2: Similar to point 1, a task definition could be desirable before describing the methodology. The defined scenarios can connect with the used downstream datasets and evaluation settings.**
>
> Thank you for the valuable suggestion. In the revision, we will add an explicit task definition that clarifies what the paper aims to accomplish. Specifically, we will define the GEO task as follows:
>
> - **Input:** a document \(d\), a query \(q\), and a Generative Engine (GE) that produces a synthesized answer with citations.
> - **Output:** a rewritten document \($\hat{d}$\) generated by a GEO model that improves the document's visibility in the GE's generated answer while preserving factual correctness and semantic fidelity.
> - **Objective:** to automatically discover GE-specific content preference rules, and use them to guide GEO models that rewrite documents so as to maximize visibility without degrading GE utility.
>
> This task definition naturally links to the downstream datasets and evaluation setting:
> - The **datasets** provide the domain-specific documents–query on which visibility is measured.
> - The **evaluation metrics** (GEO metrics and GE utility) directly operationalize the above objective.
> - The **methodology** (rule discovery and rule-guided rewriting) is then introduced as a principled solution to this defined task.
>
> We will incorporate this clarified task formulation ahead of the methodology to ensure that readers can directly connect the problem setup with our datasets and evaluation pipeline.

---

### Author Response · Authors · 2025-11-21
**General Response: Human Study on Rule Quality and Domain Alignment**

To evaluate the quality of rules extracted from generative engines, we conduct a human study with 20 annotators, all CS graduate students, who are independent of the model development process. The study is designed to measure both the clarity and validity of rules and their alignment with the intended domain.

**1. Rule Quality Assessment**

Each annotator evaluates rules independently and without knowledge of the model identity. Following prior work [1], each rule is rated on three criteria using a 1–3 scale (3 = highest quality):

- **Clarity:** whether the rule is unambiguous and easy to understand.
- **Semantic validity:** whether the rule is logically correct.
- **Decisionability:** whether a human can reliably make a binary judgment based on the rule.

We evaluate rules from the open-source **DeepSeek-R1** model (new experiment) and the closed-source **Gemini** model on the E-commerce dataset. Rules are randomly shuffled across models to avoid ordering or model bias.

**Results**
| Model | Clarity | Semantic Validity | Decisionability |
|-------|----------|-------------------|------------------|
| DeepSeek-R1 | 2.84±0.44 | 2.84±0.40 | 2.54±0.66 |
| Gemini | 2.85±0.41 | 2.84±0.41 | 2.51±0.67 |

Across both models, the average score exceeds 2.5 for all three dimensions. In particular, **95.47% of rules** are rated medium or high on all criteria, indicating that the extracted rules are generally clear, coherent, and actionable.

**2. Domain Alignment**

Annotators also categorize each rule as *E-commerce*, *Research*, or *Mixed*, independently. This evaluation is performed on the E-commerce and Research GEO datasets of Gemini generative engine.

Results show that **42.31%** of rules are correctly assigned to their specific domain, while **34.62%** fall into the *Mixed* category, reflecting general high-quality attributes such as factual accuracy and verifiability. In total, **76.93%** of rules are validated as meaningfully aligned with GE preferences rather than domain-unrelated noise.

Overall, this human study provides direct evidence that the extracted rules are rational, interpretable, and faithfully reflect the GE domain. We will include this content in the revision.

[1] Shirvani-Mahdavi, Nasim, et al. "Rule2Text: Natural Language Explanation of Logical Rules in Knowledge Graphs." International Joint Conference on Rules and Reasoning. Cham: Springer Nature Switzerland, 2025.

---

### Author Response · Authors · 2025-12-02
**Summary of Reviews, Discussions, and Paper Updates**

Dear PCs, SACs, ACs, and Reviewers,

Below is a concise summary of reviewer feedback and rebuttal discussions, highlighting resolved concerns and reviewer-specific viewpoints for quick reference.

### **1. Key Strengths**
Reviewers consistently acknowledged the importance, timeliness, and practical value of the GEO problem, as well as AutoGEO's clarity and utility:

- Problem significance: highlighted by `YwZa`, `CXWv`, and `WpqU`.
- Clear writing: emphasized by `CXWv` and `WpqU`.
- Effective system: emphasized by `YwZa`, `CXWv`, and `WpqU`.
- Comprehensive experiments: multi-domain evaluations across several SOTA LLMs, along with extensive analysis, praised by `YwZa`, `CXWv`, and `WpqU`.

Notably, **Reviewer `WpqU` confirmed an score increase**, citing fully resolved concerns. Reviewer `YwZa` remained positive throughout.

### **2. Discussion Summary**

**Reviewer `YwZa` (Rating 8 / Confidence 4)**
Requested clearer positioning relative to prior work and explicit contributions. Rebuttal clarified corpus-level rule extraction and merging, the visibility–utility objective, structured rule release, and improved related work comparisons. `YwZa` remained positive; clarity praise was echoed by `CXWv` and `WpqU`.

**Reviewer `CXWv` (Rating 4 / Confidence 2)**
Raised concerns about novelty, rule validity, and cost analysis. We provided full cost accounting, a 20-person human evaluation confirming rule clarity, and clearer explanation of contributions. We believe all concerns from `CXWv` have been addressed. Other reviewers (`YwZa`, `WpqU`) emphasized the system's significance and practicality, which partly offsets `CXWv`'s concerns regarding novelty.

**Reviewer `WpqU` (Rating 6 / Confidence 4, Confirming Score Increase)**
Requested human studies, ablations, global-optimization modeling, and factuality checks. Rebuttal added human evaluation, complete ablations, global-optimization results showing visibility dilution but utility gain, and factuality-checked case studies. `WpqU` confirmed he/she will raise the score, aligning with the positive assessments from `YwZa` and `CXWv` regarding experimental completeness.

### **3. Paper Updates**
All rebuttal clarifications, new analyses, and experiments, including human evaluation, ablations, updated cost analysis, global-adoption results, and expanded case studies, are **already included in the updated PDF and Supplementary Materials**, with revisions marked in **blue font**.

The reviewers' feedback has strengthened our work. We believe all concerns have been addressed and respectfully hope for full and fair consideration. Thank you for your thoughtful feedback and engagement.

---

### Meta-Review · Area_Chair_JXWf · 2025-12-10

**Summary:**

### Paper summary
The paper proposes AutoGEO, a framework designed to optimize web content for generative engines in a cooperative manner. It utilizes an LLM-driven pipeline to automatically discover preference rules from GEs and uses these rules to rewrite content, aiming to improve visibility.

### reviewer summary
Reviewer opinion was a little mixed: supporters find the results truly compelling, with improvements exceeding 30%. Critics argue that the technical novelty is limited, essentially combining existing prompt engineering and distillation techniques, and also point out that the rule discovery is costly. The author did supplement the cost analysis and manual evaluation in the rebuttal, but failed to address the critics' core objection regarding novelty, so the disagreement persists.

### AC comments

Even this paper lack of novelty, but the work is strong in problem importance, clarity, and practicality: it provides an interpretable preference-learning mechanism and demonstrates how those preferences can be embedded into both an immediate API system and a cost-efficient small-model alternative, supported by quantitative results and cost analysis. The remaining concerns are largely structural to the problem setting (engine drift, ecosystem externalities, and metric alignment), and do not undermine the core contribution as a systematic GEO framework.

**Reviewer Concerns:**

Addressed:

1. Cost analysis: Reviewer CXWv previously criticized the lack of cost calculation for rule extraction. The authors supplemented the cost table in their rebuttal.

2. Rule validity: Concerns were raised about the reliability of automatically generated rules. The authors supplemented a 20-person user study demonstrating that the rules are indeed clear and actionable.

Outstanding:

Lack of novelty: Regardless of additional experiments, the core remains “Direct application of existing techniques”, which cannot be altered.

**Reviewer Scores:**

Reviewer YwZa 8 likely to maintain 8. He prioritizes performance; as long as the results are good, that's sufficient.

Reviewer WpqU 6 likely to increase to 8. The rebuttal addressed some of his concerns.

Reviewer CXWv 4: will definitely maintain a 4. His core objections are “innovation” and “technical depth”—flaws that the rebuttal cannot remedy through additional experiments.

---

### Decision · Program_Chairs · 2026-01-26

Accept (Poster)